# Recombination Between Bubaline Alphaherpesvirus 1 and Bovine Alphaherpesvirus 1 as a Possible Origin of Bovine Alphaherpesvirus 5

**DOI:** 10.3390/v17020198

**Published:** 2025-01-30

**Authors:** Bruna S. Paredes-Galarza, Fabrício S. Campos, Martha T. Oliveira, Bruno A. Prandi, Ueric J. B. de Souza, Dennis M. Junqueira, Darren P. Martin, Fernando R. Spilki, Ana C. Franco, Paulo M. Roehe

**Affiliations:** 1Laboratory of Virology, Department of Microbiology, Immunology and Parasitology, Institute of Basic Health Sciences, Federal University of Rio Grande do Sul, R. Ramiro Barcelos 2600/523, Porto Alegre 90035-003, RS, Brazil; brunaparedes97@gmail.com (B.S.P.-G.); marthatoliveira@gmail.com (M.T.O.); aschidamini.b@gmail.com (B.A.P.); anafranco.ufrgs@gmail.com (A.C.F.); 2Bioinformatics and Biotechnology Laboratory, Campus of Gurupi, Federal University of Tocantins, Gurupi 77410-570, TO, Brazil; uericjose@gmail.com; 3Department of Biochemistry and Molecular Biology, Federal University of Santa Maria (UFSM), Roraima Av., 1000, Santa Maria 97105-900, RS, Brazil; dennismaletich@hotmail.com; 4Computational Biology Group, Department of Integrative Biomedical Sciences, Institute of Infectious Disease and Molecular Medicine, University of Cape Town, Anzio Road Observatory 7549, Cape Town 7700, South Africa; darrenpatrickmartin@gmail.com; 5Institute of Health Sciences, Feevale University, Campus II, RS 239, Novo Hamburgo 93525-075, RS, Brazil; fernandors@feevale.br

**Keywords:** Bubaline alphaherpesvirus, BuAHV1, BoAHV1, BoAHV5, phylogenetic analysis

## Abstract

Bovine alphaherpesvirus 1 (BoAHV1) is prevalent in cattle throughout the world, whereas bovine alphaherpesvirus 5 (BoAHV5) prevalence seems restricted to some countries in South America, Australia, and other regions, mainly in the Southern Hemisphere. BoAHV5 infections occur where water buffalo (*Bubalus bubalis*) farming is practiced, often close to cattle (*Bos taurus*) farms. Bubaline alphaherpesvirus 1 (BuAHV1), a virus whose natural host is believed to be the water buffaloes, usually causes asymptomatic infections in that species. Here, evidence is provided confirming the close relationship between BuAHV1 and BoAHV5. Phylogenetic and recombination analyses were used to reveal the evolutionary relationship between all whole-genome sequences of BoAHV1 (n = 52), BoAHV5 (n = 7), and BuAHV1 (n = 6) available to date. It is proposed here that BoAHV5 most likely resulted from multiple recombination events between a BuAHV1-like ancestor and BoAHV1-like viruses. The BoAHV5 whole unique short (US) region and most of the unique long (UL) genomic regions seem to have been derived from a BuAHV1-like parental genome, whereas at least six small segments of the UL (corresponding to nucleotides 8287 to 8624; 10,658 to 14,496; 48,013 to 48,269; 71,379 to 71,927; 81,426 to 85,003; and 94,012 to 96,841 of the BoAHV5 genome) and two small segments of the US (corresponding to nucleotides 107,039 to 107,581 and 131,267 to 131,810) have been derived from a BoAHV1-like parental genome. The hypothesis that the BoAHV5 species may have originated following a series of recombination events between BuAHV1 and BoAHV1 variants is consistent with the geographical distribution of BoAHV5, which seems to be prevalent in the regions where cattle and water buffalo farming overlap.

## 1. Introduction

Bovine alphaherpesvirus 1 (*Varicellovirus bovinealpha1*, BoAHV1) and 5 (*Varicellovirus bovinealpha5*, BoAHV5) are major pathogens of cattle. BoAHV1 subtypes (1.1, 1.2a, and 1.2b) have been associated with multiple clinical conditions including infectious bovine rhinotracheitis (IBR), vulvovaginitis, and balanoposthitis [1,2]. BoAHV5, which currently includes three defined subtypes (a, b, and c) [3], is an important cause of non-suppurative encephalitis in South American cattle and may be more prevalent than BoAHV1, at least in some regions of the continent [4,5].

Although the isolation of BoAHV1 and BoAHV5 is frequently described, reports on the isolation of Bubaline alphaherpesvirus 1 (*Varicellovirus bubalinealpha1*, BuAHV1) are uncommon. The latter was first recovered from the prepuce of asymptomatic Australian water buffaloes [6]. The pathogenic potential of this virus in water buffaloes has only partially been investigated [7,8,9,10]. On occasion, BuAHV1 usually causes subclinical infections in water buffaloes (*Bubalus bubalis*) which seem to be its natural host species [6,8,10,11]. BuAHV1 has also been associated with respiratory disease and abortion [9,12].

BoAHV1, BoAHV5, and BuAHV1 all share a noteworthy percentage of genomic identity: 76.7% between BoAHV1 and BuAHV1, 82% between BoAHV1 and BoAHV5, and 92% between BoAHV5 and BuAHV1 [13]. Due to this genomic similarity, antigenic cross-reactivity is also a common feature between these viruses. Therefore, it is plausible that wherever BuAHV1 might be detected in diagnostic laboratories around the world, it is commonly misidentified as either BoAHV1 or BoAHV5 [14,15]. The inability of serological assays to distinguish between BuAHV1, BoAHV1, and BoAHV5 infections [16,17] means that these viruses can only be accurately identified using molecular biology techniques. Even then, finding one genomic region that can be used to differentiate between the three viruses can be difficult [18]. Distinguishing between these viruses is particularly relevant in regions where buffaloes and cattle are raised nearby [8,15].

Moreover, these viruses seem capable of infecting beyond their putative hosts. For example, water buffaloes, goats, and sheep have all been successfully infected with BoAHV1 under experimental conditions [19]. Likewise, BoAHV5 has also been shown to be capable of infecting sheep and goats experimentally [5]. Furthermore, natural antibodies against BoAHV5 have been detected in sheep, although there are no reports on virus isolation in this species [20]. On the other hand, the natural susceptibility of water buffaloes for both BoAHV1 and BoAHV5 has been recently demonstrated, including one case of coinfection [21]. While natural infections with BuAHV1 in species other than its putative natural host have not been investigated and, therefore, not proved, infection of cattle has been experimentally demonstrated [22]. Thus, coinfection of individual host animals, either bovines or bubalines, with two—or more—of these closely related viruses is a possibility that should be kept in mind.

The origin of BoAHV5 and its evolutionary relationships with BoAHV1, BuAHV1, and other herpesviruses have not yet been fully examined. Previous restriction endonuclease fragment-length polymorphism-based studies [23,24] and phylogenetic studies based on partial sequences of the glycoprotein B and D genes [7,14] have indicated that BuAHV1 might be more closely related to BoAHV5 than to BoAHV1. It has also been hypothesized previously that BoAHV5 might have originated from BuAHV1 [25], although no experimental data have ever been reported to support such a hypothesis. Studies on these evolutionary relationships have been hindered, until recently, by the lack, of published BuAHV1 and BoHV5 complete genome sequences. With advances in sequencing technologies, isolates from Australia, Brazil, Italy, India, and Argentina have been sequenced [13,26,27,28,29], allowing for deeper phylogenetic analysis of such viruses. Therefore, in this study, we examine the evolutionary relationships among BoAHV1, BoAHV5, and BuAHV1, using comparative phylogenetic and recombination analyses of a set of publicly available, full-length alphaherpesvirus genomes.

## 2. Materials and Methods

### 2.1. Genomic Sequence Data from Ruminant Alphaherpesviruses

All complete genome sequences of the three ruminant alphaherpesviruses (BoAHV1, BoAHV5, and BuAHV) available in GenBank as of October 2024 were selected for analysis (Appendix A). The dataset included 52 sequences of BoAHV1, 7 of BoAHV5, and 6 of BuAHV1; for whole-genome analysis, partial genome sequences were excluded.

### 2.2. Phylogenetic Analysis

The full-length genomic sequences were aligned using the default settings of the online MAFFT version 7 [30]. The alignment was manually trimmed using the Geneious software (v.R9.1.8). The alignment was subsequently used for recombination and phylogenetic analyses to determine the evolutionary relationships between BoAHV1, BoAHV5, and BuAHV1. Additionally, complete BoAHV1, BoAHV5, and BuAHV1 gene sequences, corresponding to genes *UL*27 (which encodes glycoprotein B; gB), *UL*44 (gC), *US*6 (gD), *US*8 (gE) and *US*4 (gG), were extracted from each of the full genomes. Alignments of each gene sequence, as well as other genomic segment sequences, were performed using MAFFT alignment in Geneious.

Phylogenetic relationships were inferred by the maximum likelihood method, using the IQ-TREE v2.0.3 Database [31,32] with the setting of 1000 bootstrap replicates. The phylogenetic tree was designed in FigTree v1.4.4 [33]. Additionally, to further explore the relationships between BoAHV1, BoAHV5, and BuAHV1, maximum likelihood phylogenetic trees of partial genome sequences (putative recombinant regions) and of the genes *UL*27, *UL*44, *US*6, *US*8, and *US*4 were constructed.

### 2.3. Recombination Analyses

Recombination events, probable parental isolates of recombinants, and recombination breakpoints were detected in RDP4.101 software [34]. The alignment of the 65 genomes was analyzed in RDP4 using default settings, applying seven detection methods: RDP, GENECOV, Chimera, MaxChi, BOOTSCAN, 3Seq, and SISCAN. True recombination events were only considered credible if (i) they were detectable by five or more of these seven different methods with a *p*-value < 0.05 and (ii) they were supported by phylogenetic evidence of recombination.

To confirm the results indicated by the RDP4 analysis, and to validate the RDP4 inferred positions of recombination breakpoints, the alphaherpesvirus genome alignments were further analyzed by SimPlot 3.5.1v software [35], using the alleged recombinants as query under the following settings: window size of 200 bp, a step size of 20 bp, and Kimura (2 parameters) distance model and maximum likelihood distance model with 1000 bootstraps. Complete genome alignments of each viral group were used for analysis; specifically, the consensus of the seven BoAHV5 sequences was used as a query against the consensus of the fifty-two available BoAHV1 sequences and the six available BuAHV1 sequences.

## 3. Results

### 3.1. Phylogenetic Analysis

In order to investigate the evolutionary relationship between the three alphaherpesviruses (BuAHV1, BoAHV1, and BoAHV5), a maximum likelihood phylogenetic tree, inferred from the full-length genome sequences available in GenBank (https://www.ncbi.nlm.nih.gov/genbank/about/, accessed on 10 October 2024), was constructed. Based on the degree of pairwise genome sequence identity shared between the viruses, BoAHV5 is evolutionarily more closely related to BuAHV1 with a shared identity of 93.25% to 96.34% than to BoAHV1 (81.86% to 82.49% shared identity; Figure 1). BoAHV5 and BuAHV1 shared a direct common ancestor supported by high bootstrap values (100%) suggesting a common origin in the past. Since viral glycoproteins are important markers for selective pressure and viral evolution, as they better reflect adaptations to their hosts and differences between lineages, glycoproteins B, C, D, E, and G were chosen to perform phylogenetic analysis. Phylogenetic trees constructed from genome segments encoding the glycoproteins studied here (gB, gC, gD, gE, and gG) were all consistent with the full genome analysis in that all BoAHV5 sequences were all clearly more closely related to BuAHV1 sequences than they were to BoAHV1 sequences (Figure 2A–E). Moreover, the BoAHV5 seems to be more closely related to the Indian BuAHV1 strains, as demonstrated by the gC, gE, and gG phylogenetic trees. One exception included the gB genes from Argentinian BoAHV5 strains (MW829288, MZ420492, and MZ364295) that clustered together with BoAHV1 (Figure 2A).

### 3.2. Recombination Analysis

Recombination analysis of the complete genomes of the 65 sequences was performed. A total of 79 potential recombination sites were found among the three viruses using RDP4 software. There were true recombinant events detected in all BoAHV5 genomes (*p* < 2.79 × 10^−11^), resulting from sequences originating from both BoAHV1 and BuAHV1, suggesting that sequences with BoAHV5 origins likely come from multiple recombination events between parental BoAHV1-like viruses and parental BuAHV1-like viruses. The major backbone of the BoAHV5 genome seems to have arisen from BuAHV1 genomes, whereas BoAHV1 contributed with short segments distributed along the genome (Figure 3). Recombination events were detected in both the UL region and the US region of BoAHV5; however, only five of which were considered true recombinants in all BoAHV5 genomes: region 8287–8624 (segment B), 10,658–14,496 (segment D), 71,379–71,927 (segment H), 81,426–85,003 (segment J), and 94,012–96,841 (segment L).

True recombinant events were also detected, by six and seven algorithms, involving BoAHV1 and BoAHV5, resulting in a recombinant BoAHV5 with the regions 56,353–58,601 (segment G*) and 85,006–88,039 (segment K*) of BoAHV1 origin (Figure 3); nevertheless, these combinations were only detected in the 3 Argentinian BoAHV5 genomes (3/6 of the available BoAHV5 genomes). These sequences encode *UL*27 and a region comprising *UL*12, *UL*11, *UL*10, and part of *UL*9, respectively, and have been described before [28,36].

To confirm these recombination events based on RDP4 prediction analysis, the results were verified by Simplot (Figure 4). In this software, BoAHV1 and BuAHV1 whole-genome alignments were set as references and the BoAHV5 whole-genome alignment was set as the query. After a consensus of each viral type analysis, Simplot strongly supported the occurrence of the same recombination events detected by RPD4 (segments B, D, G, H, J, and L); however, it further considered another three regions of BoAHV5 genome as being of BoAHV1-like virus origin (segments F, N, and P). Although these regions had been detected by RDP4, they were not considered true events, due to weaker confidence by the detection methods in RDP4: segments F and N were significantly detected by only two algorithms (GENECOV and Bootscan), whereas segment P was significantly detected by only three algorithms (GENECOV, Bootscan, and 3Seq). These results are summarized in Table 1.

Both Simplot and RDP4 analyses agreed that 91.8% of the BoAHV5 genome (segments A, C, E, G, I, K, and M) was probably derived from a parental genome that was >90% identical to all BuAHV1 genome sequences currently published (Figure 2A). The five recombination events demarcate 11 different segments in all BoAHV5 genomes. These analyses also corroborate that three small *UL* region segments collectively accounting for 8.1% of the BoAHV5 genome (segments B, D, H, J, and L) were possibly derived from a parental virus closely related to BoAHV1. If the somewhat lower threshold for evidence of true recombination events (segments F, N, and P) is taken into account, it can be estimated that 91.2% of the BoAHV5 genomes are most likely derived from BuAVH1-like parental viruses, whereas 8.8% of the genomes seem derived from BoAHV1-like parental viruses.

The recombination events cover a number of BoAHV5 genes. Segment B encodes part of the coding region *UL*51 gene (palmitoylated protein cytoplasm). Segment D contains genes encoding the tegument protein (*UL*49 gene), trans-inducing factor tegument (*UL*48 gene), and the tegument phosphoprotein (*UL*47 gene). Segment F encodes the DNA polymerase catalytic subunit (*UL*30 gene). Segment H includes the major capsid protein (*UL*19 gene), and segment J involves the minor tegument protein (*UL*15), threonine protein kinase (*UL*13 gene), and the alkaline exonuclease (*UL*12 gene). Segment L encodes the virion protein (*UL*6 gene) and the primase complex (*UL*5 gene). Segment N includes BICP4 and segment P includes the alternative immediately early BICP0 (Table 2).

To verify whether the results of the recombination software could be confirmed, the evolutionary relationship of the segments prone to recombination was investigated. As shown in Figure 1, overall, the BoAVH5 genomes are more closely related to BuAHV1 genomes. Nevertheless, when the phylogenetic analysis is performed using only the potential recombinant segments, with each of these being used separately to construct a maximum likelihood phylogenetic tree, BoAVH5 always comes up more closely related to BoAHV1 (Figure 5 and Figure 6), further corroborating the results of the Simplot and RDP4 analyses.

Finally, it is noteworthy that RDP4 also detected true events of recombination occurring in specific BoAHV1 strains. A true recombinant event was detected in KY215944-India BoAHV1, involving a 1320 nt segment (77,691–79,011) between all BoAHV5, BuAHV1, and BoAHV1 genomes. Two recombinant events were also detected in the BoAHV1 MG407776-USA strain, involving the segments 103,184–114,431 and 123,602–134,254, both of which are likely of BoAHV1 origin, indicating intratype recombination. Additionally, three recombinant events were detected in the AJ004801-Switzerland strain, involving the segments 4126–126,269; 103,167–105,556; and 109,199–110,882, and also involving BoAHV1 only (data are shown in Appendix A). Recombination events are natural processes that influence herpesvirus evolution. Therefore, the occurrence of events of intratype recombination of genome segments is expected to happen.

Altogether, these data suggest that BoAHV5 is the product of recombination events between viruses closely related to BuAHV1 and BoAHV1, with a BuAHV1-like virus acting as the major parent (contributing to >50% of the nucleotides in the genome, i.e., as the “backbone” of the virus) and a BoAHV1-like virus acting as the minor parent (contributing to <50% of the nucleotides, i.e., “donating” only small fragments of the genome). The fact that all of the analyzed BoAHV5 genomes carry evidence of the same eight recombination events indicates that traces of the recombination events detected here were already present within the most recent common ancestor (MRCA) of these viruses. However, this does not prove that they were present within the MRCA of all the BoAHV5 isolates that have so far been studied: the most recent common ancestor of all known BoAHV5 genomes could have existed before the MRCA of the six BoAHV5 genomes analyzed here. This is important because, while our analyses indicate that an MRCA BuAHV1 genome is presumably the major parent of the BoAHV5 recombinants amongst all the presently available full genome sequences, it remains possible that the actual major parent is a currently uncharacterized BoAHV5 lineage that diverged from the stem of the BoAHV5 type before the MRCA of the six presently characterized BoAHV5 sequences. The two recombination events restricted to Argentinian BoAHV5 strains, on the other hand, are likely separated events that have probably arisen later, between a previously available BoAHV5 virus and another BoAHV1 virus.

## 4. Discussion

Herpesviruses have been coevolving with their hosts for millions of years [37]. As such, the evolutionary relationships of herpesvirus tend to mirror that of the host species that they infect. Accordingly, several ruminant alphaherpesviruses have been found to phylogenetically cluster together with BoAHV1 and BoAHV5 [19].

The accuracy with which the evolutionary relationships can be inferred for viruses with slowly evolving double-stranded DNA genomes is expected to be highest when full-length genomes are analyzed [38]. Relative to viruses with small RNA and single-stranded DNA genomes, however, there are far fewer full genome sequences available for double-stranded DNA viruses with genomes as large as those of the herpesviruses. Therefore, phylogenetic relationships between herpesviruses have usually been inferred using individual genes [7,14]. Nevertheless, in the last few years, with the advancement in sequencing technologies, the number of new complete genome sequences deposited in databases has increased. In this study, the evolutionary relationships of full genome sequences drawn from three ruminant-infecting herpesvirus species were determined using a whole-genome analysis approach. Phylogenetic analysis showed that BoAHV5 nucleotide sequences are more similar to that of BuAHV1 than of BoAHV1 (Figure 1), suggesting a common ancestor that existed more recently between BoAHV5 and BuAHV1 than that shared by BoAHV1 and BoAHV5. These findings are in agreement with previous studies in Italy, Argentina, Iran, and Brazil that also suggested the hypothesis of a common ancestor between BoAHV5 and BuAHV1 [8,10,21,39]. Due to the long period of coevolution, herpesviruses tend to go unnoticed or cause mild diseases in their natural hosts, causing only serious diseases when the viruses cross the species barrier [40,41]. Similar changes in virulence have been noted following host-switching in other alphaherpesviruses: for example, fatal encephalitis can occur when the B virus, also called cercopithecine alphaherpesvirus9, infects humans [42] and when the pseudorabies virus infects rabbits, cattle, and dogs, but neither of these viruses cause encephalitis in their natural hosts [43]. Most notably, BoAHV5 can display neurovirulence in cattle and cause a significant degree of neurological disease, unlike BoAHV1 and BuAHV1 in their respective hosts [10,22,44]. It is plausible that the neurovirulence of BoAHV5 in cattle may be related to it having undergone a host-shifting event from buffalo to cattle [45]. Furthermore, BoAHV5 has been detected in apparently healthy water buffaloes, and so far, it has not been associated with a clinical manifestation in these animals; on the other hand, so has BoAHV1 [15]. Still, the closer phylogenetic relationship between BuAVH1 and BoAHV5 seems to be reflected in the biology of these viruses.

Of the eight BoAHV5 recombinant segments described here (Figure 3), five segments (B, D, H, J, and L) presented strong statistical evidence only in RDP4 to be classified as a true recombination event. The other three segments (F, N, and P) presented as a real recombinant in Simplot, whereas they showed weaker evidence in RPD4, as demonstrated by only achieving statistical significance in two or three (out of seven) algorithms used (Table 1 and Figure 4). We ultimately decided to define the segments F, N, and P as true recombinants, based on the phylogenetic analysis of each segment individually (Figure 6). When each segment was compared against the equivalent BoAHV1 and BuAHV1 genomic regions, they strongly clustered with BoHAV-1, similarly to fragments B, D, H, J, and L. Thus, all the eight segments identified can be considered as true recombination events, evidenced by the grouping upon phylogenetic analysis.

RDP4 and SimPlot are both widely used programs for detecting recombination events in DNA sequences, but they have distinct characteristics in terms of approach, analysis methods, and result presentation. Recombination detection programs often use different statistical approaches to calculate P-values. RDP4, for example, implements multiple recombination detection methods, such as GENECONV, RDP, Bootscan, and MaxChi, among others. Each of these methods uses a different approach to assess the probability that a particular recombination event has occurred. This can lead to differences in the results and, consequently, in the P-values. GENECONV, for example, relies on the comparison of sequence pairs or triplets to identify potential recombination events. Bootscan uses bootstrapping simulations to calculate confidence around potential recombination events. MaxChi uses a chi-square-based analysis to compare sequence variability and detect recombination. Each method has its advantages and disadvantages, which may result in different P-values depending on the nature of the data and the recombination event. In contrast, SimPlot performs recombination detection by comparing sequences along the alignment. It calculates the genetic similarity between sequences within each window and generates graphs showing changes in similarity, where recombination events are suggested by changes in these graphs. SimPlot uses genetic distances (such as Kimura-2 distance or p-distance) to calculate the similarity between sequences in each window. Changes in similarity along the sequence suggest potential recombination events; however, it does not provide the user with P-values. Therefore, both types of software provide reliable data on recombination, based on different methods, inasmuch as both software agreed upon recombination locations, just diverging in the strength of statistical analysis.

The eight recombinant segments described here in BoAHV5 involve the *UL*51, *UL*49, *UL*48, *UL*47, *UL*30, *UL*19, *UL*15, *UL*13, *UL*12, *UL*6, *UL*5, and *BICP4* genes. These presumably BoAHV1-derived genes encode VP16, VP13/14, a tegument protein, a protein kinase, a deoxyribonuclease, and most of a helicase subunit. VP16 binds to one of the recognition elements (TAATGARAT) in the promoters of immediate early (IE) herpesvirus genes via interaction with host cellular proteins and stimulates the expression of these genes [46]. BoAHV1 and BoAHV5 sequences analyzed here encode identical residues between positions 365 and 390 of this protein suggesting that these BoAHV5 isolates plausibly express a VP16 that has the same E1 promoter transactivation activities in cattle as those of the BoAHV1 VP16. It is possible, therefore, that the recombinational acquisition by BoAHV5 of this gene from BoAHV1 could have directly impacted the replication of BoAHV5 in cattle, adding to the pathogenicity of this virus.

The evolution of alphaherpesviruses shows that the occurrence of closely related viruses and host shifts (spillovers) favor the emergence of recombinant viruses that could infect different species [37]. Recombination has been detected previously in alphaherpesviruses [19,47,48]. Interspecies recombination events have been reported both in vitro and in vivo between human alphaherpesviruses 1 and 2 [49]. Recombination events have also been reported between equine alphaherpesviruses (EqAHVs), particularly between EqAHV1 and EqAHV4, and between EqAHV4 and EqAHV9 [50]. In cattle, the first evidence of in vitro interspecies recombination between BoAHV1 and BoAHV5 was reported by Maurens et al. [51]. The first evidence of natural recombination between two species of bovine alphaherpesviruses was reported by Maidana et al. [36]. In that study, three field BoAHV5 isolates (MW829288, MZ420492, and MZ364295) were identified as recombinants between BoAHV1 and BoAHV5; additionally, two recombination hotspots were detected in the *UL*27 gene, which encodes gB. These findings suggested that the BoAHV5b subtype should be considered a natural recombinant between BoAHV1.2b and BoAHV5 [36]. Recently, ref. [28] obtained the complete genomes of these natural recombinants, and the analyses showed a second recombination event in two of the three viruses in the segment encoding the *UL*11, the *UL*10, and part of the *UL*9 genes. This new recombination seems to have originated independently from the first recombination event involving gB [28]. In those studies, co-circulation and coinfection of related viruses within a herd or country seem to be prerequisites for recombination to take place. The results presented here corroborate the results of those authors; both recombination events were detected here (shown in Figure 3 as segments G* and K*) in our analysis. The fact that those particular recombination events were restricted to Argentinian BoAHV5 strains could suggest a more recent event in bovine herpesvirus evolution.

One of the most peculiar aspects of the biology of BoAHV1, BoAHV5, and BuAHV1 is their relative geographical distributions. While BoAHV1 is found in cattle populations worldwide, the geographical range of BoAHV5 is more restricted. Besides isolated reports of BoAHV5 in Italy [52], Australia [53], Canada [54], Hungary [55], Iran [56], India [57] and the USA [58], older records have not been confirmed by sequencing, so it is not possible to confirm whether those were in fact BoAHV1 or BoAHV5. The virus has only been frequently and reliably identified in South America [4,5,59,60,61,62]. It has been speculated that vaccination against BoAHV1 might be responsible for the absence of BoAHV5 in cattle in the Northern Hemisphere [58,63], yet no experimental data have been produced to confirm such a hypothesis. The bovine family (*taurids* and *bisonids*) diverged from a common ancestor to generate the water buffalo (*Bubalus bubalis*) and the African buffalo (*Syncerus caffer*) lineages between five and ten million years ago [64]. Thus, it might be conjectured that the most recent common ancestors of BuAHV1 and both BoAHV1 and BoAHV5 possibly already existed more than 5 million years ago. Cattle, however, were domesticated approximately 10,500 years ago [65] in the Middle East and the Indian subcontinent and were introduced into Europe in 6400 BCE [66].

The apparent origin of the seven analyzed BoAHV5 genomes that likely took place through recombination events between BuAHV1 and BoAHV1 would imply that, at some time over the past 10,500 years, buffaloes and cattle must have shared the same natural environment—possibly in a region where the geographical distribution of buffaloes infected with BuAHV1 and cattle infected with BoAHV1 would overlap. That common ancestor of the analyzed BoAHV5 genomes probably arose during this period, following an infection of an individual bovine or buffalo with both a BuAHV1-like and BoAHV1-like ancestor. However, the data presented here are also consistent with another hypothesis: BoAHV5 would be the offspring of a divergent (presently uncharacterized) non-recombinant BuAHV1 lineage. Whereas the first hypothesis implies that the entire BoAHV5 lineage could have arisen following a series of recombination events that occurred at any time over the past 10,500 years, the second implies that the BoAHV5 lineage arose through the gradual accumulation of point mutations on a divergent BuAHV1-like virus; then, a relatively recent series of recombination events between the divergent BuAHV1-like and BoAHV1 genomes yielded the MRCA of the seven BoAHV5 genomes. The fact that the BoAHV1-like segments that are present within these BoAHV5 genomes cluster phylogenetically among the currently sequenced BoAHV1 genomes (Figure 5 and Figure 6) suggests a recent origin for these recombination events and is, therefore, more consistent with the second hypothesis than the first.

## 5. Conclusions

In summary, the present study examined the evolutionary relationships of BuAHV1, BoAHV1, and BoAHV5 using all publicly available full genome sequence data on these three alphaherpesvirus species. These investigations revealed that the MRCA of the seven analyzed BoAHV5 genomes was more closely related to BuAHV1 than to BoAHV1. Currently, BoAHV5 and BuHV1 genomes share a nucleotide identity of 92%, while BoAHV5 and BoAHV1 genomes share only 82% of sequence identity. Although not the only possibility, the data presented here support the hypothesis that the MRCA of BoAVH5 originated from recombination events between BuAHV1-like and BoAHV1-like ancestors, where BuAHV1 provided the major parent (donor) genome. A definitive answer on the origins and the moment in time of the recombination events in BoAHV5 will require sequencing of more BoAHV5 and BuAHV isolates so that it can be determined whether there is, in fact, an MRCA for all presently circulating BoAHV5. If this is true, then it would imply that recombination played a defining role in the origin of the BoAHV5 species and likely occurred before the modern agricultural era. Alternatively, the recombination events may have emerged relatively recently in the evolutionary history of BoAHV5, and the emergence and spread of these recombinants could therefore potentially be attributable to recent human-mediated changes in the epidemiology of bovine herpesviruses.

## Figures and Tables

**Figure 1 viruses-17-00198-f001:**
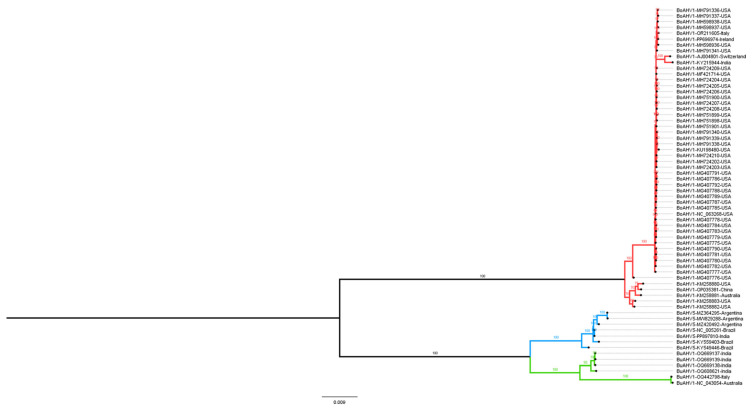
Phylogenetic relationship between sixty-five bovine and bubaline alphaherpesvirus complete genomes. Each sequence is shown as a black dot and is followed by its name, the GenBank accession number, and country of origin. Red lines indicate BoAHV1; blue lines, BoAHV5; and green lines, BuAHV1. Phylogenetic analysis was performed using the maximum likelihood method in the IQ-TREE v2.0.3 Database with the settings for 1000 bootstrap replicates. Numbers associated with particular branches indicate the percentage of 1000 bootstraps.

**Figure 2 viruses-17-00198-f002:**
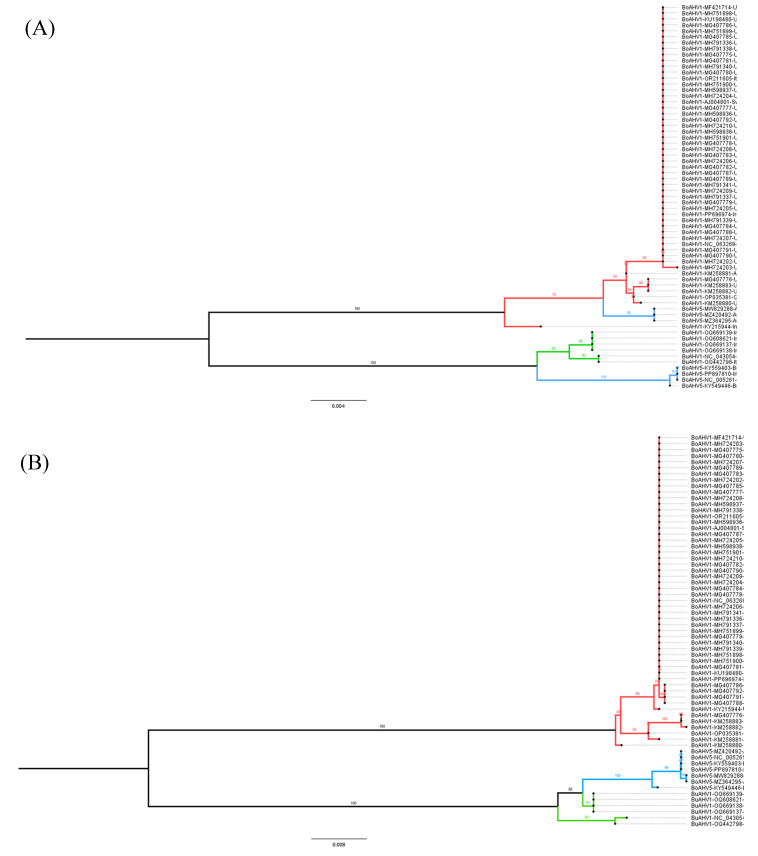
Phylogenetic analysis of BoAHV1, BoAHV5, and BuAHV1 glycoproteins. Alignments were performed with complete gene sequences extracted from full-length genome sequences using MAFFT. Red lines indicate BoAHV1; blue lines, BoAHV5; and green lines, BuAHV1. Phylogenetic analysis was performed using the maximum likelihood method in the IQ-TREE v2.0.3 Database with the settings for 1000 bootstrap replicates. Numbers associated with particular branches indicate the percentage of 1000 bootstraps. All glycoprotein genes analyzed from BoAHV5 were more closely related to BuAHV1 genes than BoAHV1, with the exception of three gB BoAHV5 from Argentinian strains. (**A**) Glycoprotein B (*UL27* gene); (**B**) glycoprotein C (*UL44* gene); (**C**) glycoprotein D (*US6* gene); (**D**) glycoprotein E (*US8* gene); and (**E**) glycoprotein G (*US4* gene).

**Figure 3 viruses-17-00198-f003:**
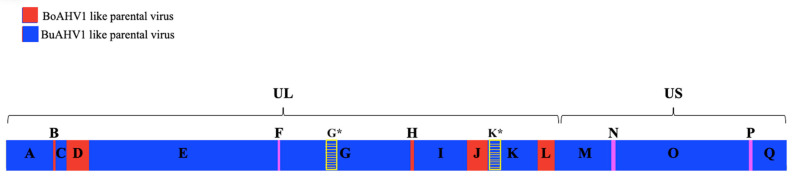
Proposed evolutionary origins of the BoAHV5 genome. The unique long (*UL*) and unique short (*US*) regions are indicated. Recombinant segments detected by both RDP4 and Simplot, derived from a BoAHV1-like parental virus, are identified by red blocks (segments A-E, G-M, O, and Q) or by pink blocks when recombinant segments show significance in less than 5 algorithms by RDP4 (segments F, N, and P); hatched yellow blocks represent recombinant events between BoAHV1-like and BoAHV-5 parental viruses detected only in the genome of Argentinian BoAHV5 strains (segments G* and K*). The blue regions represent the segments derived from a BuAHV1-like parental virus. The sizes of the different regions are drawn to scale.

**Figure 4 viruses-17-00198-f004:**
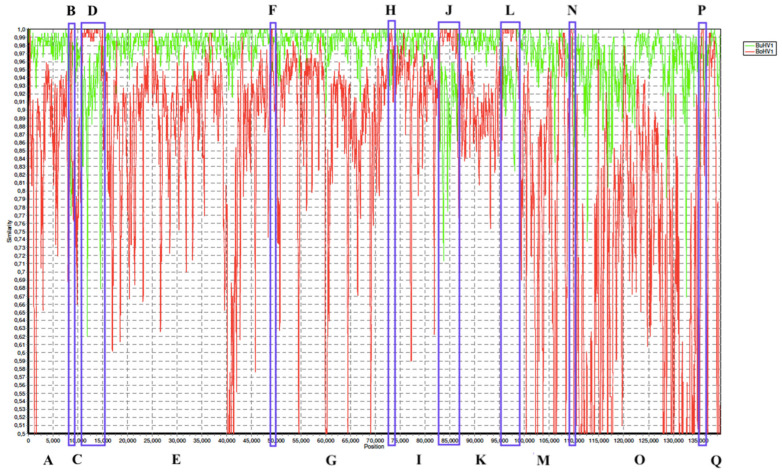
SimPlot++ analysis of consensus sequence identities shared between BoAHV5, BuAHV1, and BoAHV1. The relative consensus genetic distances between BoAHV5 and both BoAHV1 (red line) and BuAHV1 (green line) were compared across the entire length of the BoAHV5 genome. Overall, BoAHV5 is more closely related to BuAVH1; however, there are genome regions where an inversion of identity can be observed. These are considered recombination event regions (blue blocks, indicated with letters). The Y-axis indicates the shared identity percentage to the BoAHV5 genome within a sliding window 200 bp wide, moved along the genome in 20 bp increments, centered on the plotted position. The X-axis represents the position on the BoAHV5 genome alignment. Analysis was performed using the Kimura (2 parameters) distance model and maximum likelihood model with 1000 bootstrap replicates.

**Figure 5 viruses-17-00198-f005:**
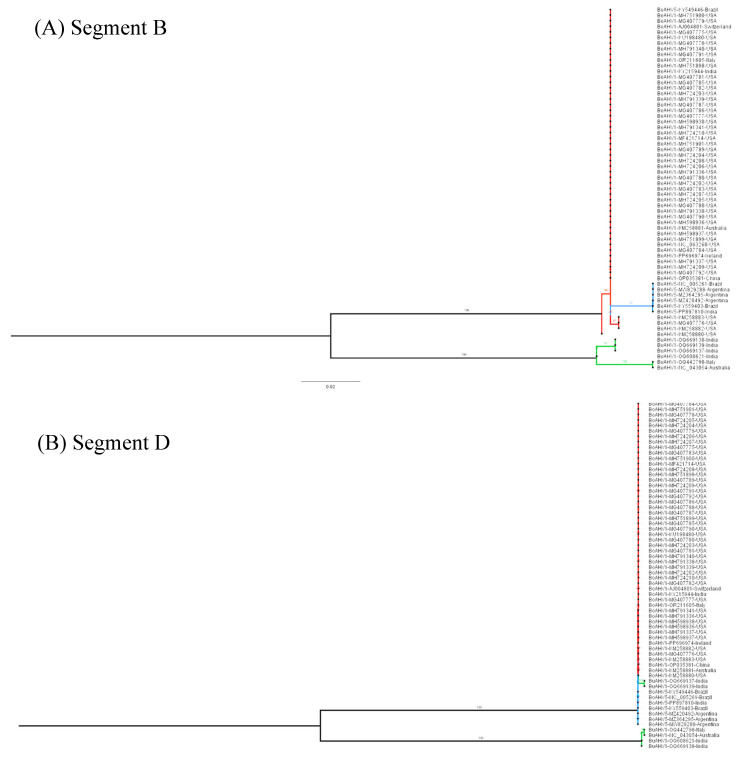
Maximum likelihood phylogenetic trees constructed from five true events detected by RDP4 and Simplot software. The BoAHV5 segments detected as true events by the recombination software showed clustering with BoAHV1 genome sequences. Phylogenetic analysis of (**A**) segment B (*UL*51), (**B**) segment D (*UL*47, 48, 49), (**C**) segment H (*UL*19), (**D**) segment J (*UL*12,13,15), and (**E**) segment L (*UL*5, 6). Phylogenetic analysis was performed using the maximum likelihood method in the IQ-TREE v2.0.3 Database with the setting of 1000 bootstrap replicates. Numbers associated with particular branches indicate the percentage of 1000 bootstraps.

**Figure 6 viruses-17-00198-f006:**
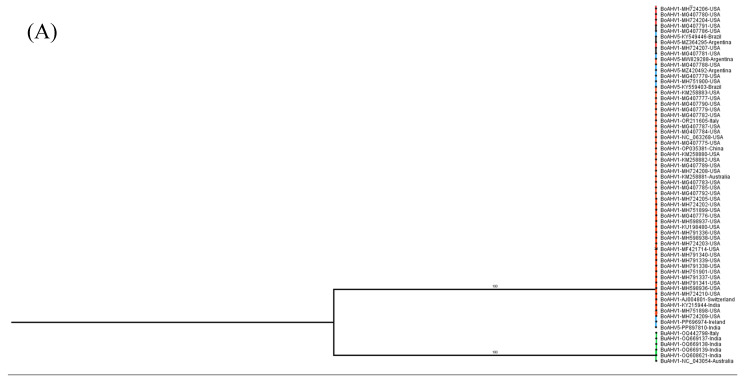
Phylogenetic analysis based on (**A**) segment F (*UL*30); (**B**) segment N (BICP4); (**C**) segment P (BICP4); (**D**) segment G* (*UL*27); and (**E**) segment K* (*UL*12, *UL*11, *UL*10, and *UL*9) of BoAHV1, BoAHV5, and BuAHV1 genomes. Phylogenetic analysis was performed using the maximum likelihood method in the IQ-TREE v2.0.3 Database with the setting of 1000 bootstrap replicates. Numbers associated with particular branches indicate the percentage of 1000 bootstraps.

**Table 1 viruses-17-00198-t001:** Summary of unique recombination events identified by RDP4.

Segment	Event	Breakpoint				Recombinant Sequence	Minor Parental Virus	Major Parental Virus	Detection Method						
		In Recombinant Sequence		In Alignment					RDP	GENECONV	Bootscan	Maxchi	Chimera	SiSscan	3Seq
		Begin	End	Begin	End										
B	9	8287	8624	8449	8786	All BoAHV5	All BoAHV1	All BuAHV1	4.03 × 10^−64^	3.43 × 10^−63^	1.63 × 10^−61^	3.18 × 10^−11^	2.79 × 10^−11^	NS ^1^	5.98 × 10^−11^
D	2	10,658	14,496	10,862	14,710	All BoAHV5	BoAHV1 (42)	All BuAHV1	7.01 × 10^−320^	5.83 × 10^−303^	9.77 × 10^−297^	1.84 × 10^−38^	9.41 × 10^−32^	7.60 × 10^−46^	6.30 × 10^−11^
F **	36	48,013	48,269	49,038	49,294	All BoAHV5	BoAHV1 (39)	All BuAHV1	NS ^1^	6.12 × 10^−8^	1.69 × 10^−6^	NS ^1^	NS ^1^	NS ^1^	NS ^1^
H	13	71,379	71,927	72,786	73,352	All BoAHV5	BoAHV1 (51)	All BuAHV1	5.10 × 10^−31^	3.34 × 10^−31^	3.24 × 10^−31^	9.10 × 10^−5^	7.97 × 10^−5^	NS	6.14 × 10^−11^
J	1	81,426	85,003	82,943	86,548	All BoAHV5	All BoAHV1	All BuAHV1	7.02 × 10^−308^	1.45 × 10^−303^	5.31 × 10^−280^	8.91 × 10^−54^	6.87 × 10^−41^	1.37 × 10^−65^	1.57 × 10^−12^
L	5	94,012	96,841	95,610	98,447	All BoAHV5	All BoAHV1	All BuAHV1	1.21 × 10^−209^	6.12 × 10^−206^	3.91 × 10^−211^	8.86 × 10^−29^	2.08 × 10^−29^	1.05 × 10^−35^	6.30 × 10^−11^
G *	4	56,353	58,601	57,476	59,751	BoAHV5 (3)	All BoAHV1	All BuAHV1 + BoAHV5 (3)	6.54 × 10^−231^	1.01 × 10^−234^	2.85 × 10^−227^	3.08 × 10^−22^	4.21 × 10^−22^	1.66 × 10^−13^	6.77 × 10^−11^
K *	3	85,006	88,039	86,549	89,638	BoAHV5 (2)	All BoAHV1	BuAHV1 (4) + BoAHV5 (5)	NS ^1^	1.50 × 10^−303^	2.00 × 10^−203^	1.21 × 10^−39^	3.30 × 10^−39^	1.05 × 10^−37^	6.77 × 10^−11^
N **	35	107,039	107,581	109,168	109,716	All BoAHV5	BoAHV1 (50)	BuAHV1 (4)	NS ^1^	4.78 × 10^−9^	1.25 × 10^−8^	NS ^1^	NS ^1^	NS ^1^	NS ^1^
P **	32	131,267	131,819	135,610	136,156	All BoAHV5	BoAHV1 (51)	BuAHV1 (4)	NS ^1^	2.97 × 10^−10^	6.96 × 10^−10^	NS ^1^	NS ^1^	NS ^1^	3.72 × 10^−2^

* True recombinant events by RDP4 detected only in Argentinian BoAHV-5 strains (events not detected by Simplot). ** Recombinant events with weaker evidence in RDP4, detected as true events by Simplot. ^1^ NS: No significant *p*-value was recorded for this recombination event using this method.

**Table 2 viruses-17-00198-t002:** Genes encoded by recombinant segments.

Segment	Gene	Known Function
B	UL51	Palmitoylated protein cytoplasm
D	UL47, UL 48, UL49	Tegument phosphoprotein, trans-inducing factor (tegument), tegument protein
F	UL 30	DNA polymerase catalytic subunit
H	UL19	Major capsid protein
J	UL12, UL13, UL15	Alkaline exonuclease, virion serine/threonine protein kinase, minor tegument protein
L	UL5, UL6	Component of DNA helicase/primase complex, virion protein
N	BICP4	BICP4 positive and negative regulator gene
P	BICP4	Alternative immediately early BICP0

## Data Availability

The sequences in this study have been deposited in NCBI GenBank.

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
