# Peer review of "Recombination Between Bubaline Alphaherpesvirus 1 and Bovine Alphaherpesvirus 1 as a Possible Origin of Bovine Alphaherpesvirus 5"

_viruses, 2025, doi:10.3390/v17020198_

Round 1
Reviewer 1 Report
Comments and Suggestions for Authors
The study in the manuscript by Paredes-Galarza et al., investigates the evolutionary origins of BoAHV5 hypothesizing it emerged from recombination between bubaline alphaherpesvirus 1 (BuAHV1) and bovine alphaherpesvirus 1 (BoAHV1). Using phylogenetic and recombination analyses of full-length genomes, the authors identified multiple recombination events. They propose that BuAHV1 acted as the major parental genome, while BoAHV1 contributed smaller segments. These findings shed light on the evolutionary dynamics of alphaherpesviruses in regions where cattle and water buffalo farming overlap.
The following comments should be addressed:
1- The manuscript relies heavily on nucleotide sequences for phylogenetic analysis. Why were amino acid sequences of key glycoproteins (e.g., gB, gC, gD) not analyzed? Amino acid-based trees might offer complementary insights and validate findings.
2- The rationale for focusing on certain genome segments over others is unclear. The authors should explain why these specific partial sequences were prioritized for phylogenetic analysis, especially within glycoprotein genes.
3- The manuscript provides exhaustive data that can overwhelm readers.
Certain tables (e.g., Table 1 listing sequences used in this study) could be moved to the supplementary section. This will streamline the manuscript and focus the attention of readers on critical results.
4- The authors analyzed 6 BoAHV1.2 sequences out of 52 BoAHV1 sequences without specifying whether they were BoAHV1.2a or BoAHV1.2b. Additionally, they did not identify which BoAHV1 subtypes served as ancestral parental genomes, which limits the interpretation of recombination events and the evolutionary origins of BoAHV5.
Reviewer 2 Report
Comments and Suggestions for Authors
The authors described possible origin of BoAHV5, which was the product of recombination events between viruses closely related to BuAHV1 and BoAHV1, through phylogenetic and recombination analyses using whole genome sequence of 65 viruses. This is a carefully done study and the findings are of considerable interest, coinciding with the previous reports. A few revisions are listed below.
1. Line 144. BuAHV1, BoAHV1, and BoAHV5.
2. Table 2. What is NS?
3. Line 342. Delete (CeAHV9) and (SuAHV1).
4. Lines 347 and 403. Cattle should be better than bovine.
5. Line 401. Delete (HuAHV1 and HuAHV2).
6. Lines 405, 406, and 411. The citations should be shown by the authors.
7. Lines 423-425. This phrase should be combined to another sentence.
8. Line 467. BuAHV1 not BuAHV.
